# Nicotine Dependence in a Banned Market: Biomarker Evidence from E-Cigarette Users in São Paulo, Brazil

**DOI:** 10.3390/ijerph22060960

**Published:** 2025-06-19

**Authors:** Jaqueline Ribeiro Scholz, Elaine Cristine D’Amico, Juliana Takitane, Daniele Mayumi Sinagawa, João Mauricio Castaldelli-Maia, Marcelo Filonzi dos Santos, Rodrigo Alves de Oliveira, Guilherme Vinicius Marques, Eric Nagamine Lima, Diana Fernanda Lasso Rodriguez, Sara Ziotti, Vilma Leyton, Maria Cristina Megid

**Affiliations:** 1Serviço de Prevencao e Reabilitacao, Instituto do Coracao, Hospital das Clinicas HCFMUSP, Faculdade de Medicina, Universidade de Sao Paulo, Sao Paulo 05508-090, Brazil; sara.ziotti@hc.fm.usp.br; 2Secretaria de Estado da Saúde, Vigilância Sanitária Estadual, Sao Paulo 01246-901, Brazil; edamico@cvs.saude.sp.gov.br (E.C.D.); mmegid@cvs.saude.sp.gov.br (M.C.M.); 3Departamento de Medicina Legal, Bioetica, Medicina do Trabalho e Medicina Fisica e Reabilitacao—LIM40, Faculdade de Medicina FMUSP, Universidade de Sao Paulo, Sao Paulo 05508-090, Brazil; julianatakitane@gmail.com (J.T.); danimayumi@usp.br (D.M.S.); msfilonzi@gmail.com (M.F.d.S.); rodrigoalves08@hotmail.com (R.A.d.O.); vileyton@usp.br (V.L.); 4Departamento de Psiquiatria, Faculdade de Medicina FMUSP, Universidade de Sao Paulo, Sao Paulo 05508-090, Brazil; jmcmaia@usp.br; 5Faculdade de Medicina FMUSP, Universidade de São Paulo, Sao Paulo 05508-090, Brazil; guilherme.marques@fm.usp.br (G.V.M.); eric.nagamine@fm.usp.br (E.N.L.); dianisfer5000@hotmail.com (D.F.L.R.)

**Keywords:** electronic cigarettes, nicotine or derivatives, toxicology

## Abstract

Although electronic cigarettes have been banned in Brazil since 2009, their use is increasing, particularly among youth. We conducted a biomarker-based study to profile exclusive e-cigarette users in São Paulo and to examine the associations of e-cigarette use with salivary nicotine and cotinine levels. A population-based, cross-sectional study was conducted between April and September 2024 in six municipalities in São Paulo, Brazil. Randomly selected participants who reported exclusive use of electronic cigarettes completed a questionnaire and provided oral fluid samples for the determination of their nicotine and cotinine concentrations using LC-MS/MS. The cohort consisted of N = 417 participants. Significant associations were found between nicotine and cotinine concentrations and variables such as knowledge of nicotine content and product type. Addiction status significantly influenced the nicotine and cotinine concentrations, as well as smoking history, last consumption, recharge/purchase frequency, and consumption duration (all *p* < 0.001). Participants who perceived themselves to have a moderate or severe addiction exhibited higher nicotine and cotinine concentrations compared with those who did not perceive that they had an addiction (*p* < 0.001). Most participants were young, predominantly White, and highly educated and earned higher incomes. The findings reveal a correlation between perceived nicotine dependence and salivary nicotine and cotinine concentrations, underscoring the physiological and behavioral markers of electronic cigarette addiction. High salivary concentrations of nicotine appear to be independent of duration of e-cigarette use, smoking history, and age. These findings underscore the urgent need for surveillance and public health interventions, even in jurisdictions where these products remain illegal. The study limitations include its cross-sectional design and potential selection bias due to convenience sampling.

## 1. Introduction

The characterization of electronic cigarette (e-cigarette) users in São Paulo, Brazil, requires a comprehensive analysis of sociodemographic characteristics, usage patterns, motivations, and potential health impacts. In Brazil, the sale and marketing of e-cigarettes has been prohibited since 2009. Nevertheless, the use of these products continues to rise, especially among youth and young adults. This regulatory paradox highlights the urgent need to better understand e-cigarette use patterns and its associated health risks in this unregulated market. Although the commercialization of these products has been prohibited in the country since 2009 by the National Health Surveillance Agency (ANVISA, Ministry of Health, Brazil) [1], a perceived increase in consumption was observed following the COVID-19 pandemic, particularly among young adults. This trend was captured by the Vigilância de Fatores de Risco e Proteção para Doenças Crônicas por Inquérito Telefônico (VIGITEL; Surveillance System for Risk and Protective Factors for Chronic Diseases by Telephone Survey), an annual survey conducted since 2006 that includes all Brazilian states; the prevalence of e-cigarette use increased from 2.3% in 2019 to 2.5% in 2020 [2]. After this, an increase was reported in young adults seeking treatment for nicotine dependence at specialized smoking cessation centers in São Paulo. The small sample, comprising 26 exclusive e-cigarette users, revealed that all participants reported a high level of nicotine dependence. This was evidenced by urinary cotinine levels of up to 600 ng/mL in semiquantitative tests, while their carbon monoxide concentrations in exhaled air were below 3 ppm [3].

Scant information exists on nicotine dependence in e-cigarette users not seeking smoking cessation treatment. Thus, analysis examining nicotine consumption biomarkers in e-cigarette users is essential for informing public health policies and intervention strategies aimed at curbing the use of such devices. The correlation between user characteristics and nicotine concentration and that of its primary metabolite—cotinine—can offer valuable insights into their usage behaviors and associated health risks, including nicotine dependence and acute toxicity [4].

High cotinine concentrations are strongly associated with higher nicotine dependence and greater challenges regarding smoking cessation [5]. Furthermore, exceptionally high nicotine concentrations may indicate hazardous usage patterns or unintentional high-dose exposure, especially among young adults; this population is particularly vulnerable to addiction and frequently exhibits biochemical markers indicative of problematic use, even when self-reported consumption is described as “casual” [6].

The aim of this study was to characterize exclusive e-cigarette users in terms of user sociodemographic data, e-cigarette consumption patterns, perception of addiction and health risks, device accessibility, health and environmental impact, government regulations, attempts to quit vaping, future perspectives, and social implications of e-cigarette use and to correlate these variables with nicotine and cotinine concentrations in oral fluid samples.

## 2. Materials and Methods

### 2.1. Study Design and Procedures

This study used a population-based, cross-sectional design to characterize e-cigarette users in six different municipalities (São Paulo, Santo Andre, Campos do Jordão, Ribeirão Preto, Campinas, and Santos) in the state of São Paulo, Brazil. Participants aged at least 18 years and reporting exclusive e-cigarette use were recruited from April to September 2024. The exclusion criteria included smoking tobacco (to eliminate dual users) or non-tobacco products (such as marijuana) and having a carbon monoxide concentration in exhaled air of 4 ppm or higher [7]. The exclusion of dual users was based on participants’ self-report of any tobacco or marijuana use, complemented by CO measurement to detect possible cigarette or combusted marijuana consumption. Those who reported dual use or presented CO ≥4 ppm were excluded. Participants who reported using other forms of nicotine, such as nicotine replacement therapy (e.g., gum and patches), were also excluded.

Consent for study participation was given by signing an informed consent form approved by the Local Ethics Committee of the Hospital das Clínicas da Faculdade de Medicina da Universidade de São Paulo (Opinion #6.775.610). The data underlying this article are available in this article and its online Appendix A.

The field data were collected in diverse public settings—including nightlife venues, gyms, universities, and workplaces—by trained health surveillance staff. Site selection was based on the official inspection schedule provided by the Health Surveillance Center of the State Department of Health of São Paulo. As such, these locations were already scheduled to receive technical inspections regardless of this research, which characterizes the sample as a random selection. This strategy was designed to reflect real-world exposure and access patterns among active e-cigarette users. Data collection included a self-administered questionnaire, the measurement of carbon monoxide concentration in exhaled air, and oral fluid sample collection using Salivette [8] for the determination of nicotine and cotinine concentrations. The samples were kept at 4 °C during shipping to the Laboratory of Toxicology at the University of São Paulo Medical School, then frozen at −20 °C and thawed prior to analysis, which was carried out no more than 10 days after sampling to ensure analyte stability [9].

To explore whether prior tobacco smoking history influenced nicotine exposure, product use patterns, or perceived addiction, participants who reported exclusive e-cigarette use were stratified into two groups: never smokers and former smokers. “Never smokers” were defined as individuals who reported never having smoked tobacco products in their lifetime. “Former smokers” were defined as individuals who reported having quit smoking tobacco products at least one month prior to the study, with exhaled carbon monoxide levels ≤3 ppm confirming abstinence. This comparison aimed to assess potential differences in behavioral and biomarker outcomes based on previous nicotine exposure, providing insights into how past tobacco use may affect current patterns of e-cigarette consumption and dependence.

### 2.2. Questionnaire Information

The questionnaire was developed by the research team based on prior national surveys and peer-reviewed literature. It has not undergone formal psychometric validation, and this limitation is addressed in the Discussion section. The information gathered by the questionnaire included sociodemographic profile (gender, age, race, years spent in education, and monthly earning income); weight and height—body mass index (BMI); physical and mental health conditions; and smoking history (former and never smokers). Regarding e-cigarette use, the participants were inquired about how they were first introduced to the product (introduction to e-cigarettes), their reasons for using it (reasons for use), what types of device they use for vaping (product type), for how long they have been using it (consumption duration), when they last used it prior to the survey (last consumption), where or how they usually acquire a device (place of purchase), how frequently they recharge or buy it (recharge/purchase frequency), as well as how much money they spend on it (monthly expenses). The participants were also asked if they are aware of the presence of nicotine in their e-cigarettes (knowledge of nicotine content), the form of nicotine present in the device (nicotine form), and its concentration (nicotine salt/free-base concentration). Data on the participants’ perception of the health impact (perception of impact on health status), the risks associated with consumption (knowledge of risks) and exposure (knowledge of passive exposure), their perception of the intensity of nicotine dependence (perception of addiction)—self-rated by participants as none, mild, moderate, severe, or unsure—and the impact on social relations (social impact) were also collected. Finally, the participants were asked whether they put themselves at risk in order to acquire e-cigarettes (exposure to risk), if they had tried to stop using them (attempts to quit vaping), whether they intended to continue using them (future perspectives on e-cigarette use), and how they felt about the prohibition of e-cigarette commercialization in Brazil (opinion on government regulations).

### 2.3. Nicotine and Cotinine Determination in Oral Fluid

Nicotine and cotinine concentrations were measured via liquid chromatography–tandem mass spectrometry (LC-MS/MS). The method was fully developed at the Laboratory of Toxicology, University of São Paulo Medical School, and although it has not yet been published, its validation adhered to standard practices for analytical method validation in forensic toxicology [10]. For sample preparation, 100 µL of oral fluid was added to 400 µL of methanol plus the cotinine-D3 Internal Standard (Cerilliant Corporation, Round Rock, TX, USA), with a final concentration of 10 ng/mL. The samples were mixed for 30 s, followed by centrifugation at 2000× *g* for 10 min. The supernatant was filtered using a 22 µm polytetrafluoroethylene hydrophobic membrane, and 20 µL was injected into the LC-MS/MS system (Shimadzu Corporation, Kyoto, Japan) with quadrupoles operating in Multiple Reaction Monitoring mode. For the separation of nicotine, cotinine, and cotinine-D3, a Shim-pack Velox Biphenyl (2.7 μm 2.1 × 100 mm) column was used (Shimadzu Corporation, Kyoto, Japan). The analyte concentration was calculated using calibration curves ranging from 5 to 2000 ng/mL and from 1 to 2000 ng/mL for nicotine and cotinine, respectively. The limit of detection (LOD) was 1 ng/mL for both nicotine and cotinine, while the limit of quantification (LOQ) was 5 ng/mL for nicotine and 1 ng/mL for cotinine. To ensure the reliability of the results, the dilution integrity was validated for 5000, 10,000, and 15,000 ng/mL. Further information on method development and validation is available in the Appendix A.

### 2.4. Data Analysis and Statistics

First, descriptive analyses were conducted. Absolute and relative frequencies were presented for the categorical variables, while summary measures (mean, quartiles, minimum, maximum, standard error, and standard deviation) were used for the numerical variables.

Mean comparisons between groups were performed using the Kruskal–Wallis test and, given no normal distribution of data, verified using the Kolmogorov–Smirnov test. When significant differences in means were identified in the Kruskal–Wallis test, pairwise group comparisons were conducted using the Dunn–Bonferroni method to maintain a global significance level. For categorical variables, the *p*-values correspond to the statistical significance of the descriptive statistics obtained from the chi-square test and Fisher’s exact test. For numerical variables, the Mann–Whitney test was used to compare non-normally distributed variables between groups, while the *t*-test was applied to normally distributed numerical variables.

To simultaneously assess the effects of demographics, clinical characteristics, and smoking history (explanatory variables) on each dependent variable (cotinine and nicotine), univariate and multivariate linear regression analyses were performed. Initially, all predictor variables were included in the model. Later, non-significant variables at the 5% significance level were removed sequentially using the backward elimination method.

Missing data were addressed by excluding cases where specific information was unavailable, as the absence of data could compromise the analysis.

A 5% significance level was adopted for all statistical tests. The analyses were conducted using the statistical software packages SPSS 20.0 (IBM Corp., Armonk, NY, USA) and STATA 17 (StataCorp LLC, College Station, TX, USA).

## 3. Results

### 3.1. Screening

Initially, 9099 individuals were approached and invited to participate in the study, and the final cohort comprised N = 417. The selection process and applied exclusion criteria are detailed in Figure 1.

### 3.2. Sociodemographic Characteristics and Smoking History

Table 1 presents the participants’ sociodemographic profiles, physical characteristics, and health conditions according to smoking history.

Significant differences were neither observed between men and women nor with respect to body mass index. The predominant population consisted of White participants under 25 years old, with high educational levels and high monthly earning incomes. Clinical conditions and mental disorders were reported by less than 32% of participants included in the analyses. Among the participants affected by psychiatric disorders, anxiety was prevalent.

Based on smoking history, distinct distributions (*p* < 0.05) were observed for gender, age, and presence of asthma. Former smokers were mostly males and significantly older than never smokers and had a higher incidence of asthma.

### 3.3. E-Cigarette Consumption and Biomarker Concentrations

Table 2 presents information on the e-cigarette consumption patterns, product characteristics, purchases and expenses, and nicotine and cotinine concentrations according to smoking history.

Based on smoking history, distinct distributions (*p* < 0.05) were observed for the following parameters: introduction to e-cigarettes (self-interest), reasons for use (out of curiosity, influence of friends and/or family, tobacco smoking cessation, and alternative to traditional cigarettes), consumption duration, place of purchase (abroad), recharge/purchase frequency, monthly expenses, nicotine form (nicotine free-base), and mean nicotine and cotinine concentrations.

Higher proportions of former smokers (compared with never smokers) were introduced to e-cigarettes due to their own interests and research on the subject and currently used the product as part of their smoking cessation treatment or as an alternative to traditional cigarettes; they purchased their devices abroad on a weekly basis, with monthly expenses up to BRL 500.00. Additionally, former smokers used more nicotine free-base products when compared with never smokers. The mean nicotine and cotinine concentrations were higher in former smokers than in never smokers.

Conversely, higher proportions of never smokers (compared with former smokers) use e-cigarettes out of curiosity or due to the influence of friends and/or family and have been vaping for up to 1 year, with occasional recharges/purchases. Much more never smokers than former smokers spend up to BRL 100.00 on e-cigarettes per month.

No statistical differences were observed between smoking history groups regarding type of e-cigarette device, duration since last consumption, knowledge of nicotine content, and nicotine salt/free-base concentrations in the e-cigarettes.

### 3.4. Nicotine and Cotinine Concentrations in Oral Fluid

The nicotine and cotinine concentrations detected in the saliva of former and never smokers are presented with respect to the duration since last e-cigarette consumption (Figure 2) and e-cigarette recharge/purchase frequency (Figure 3).

In the multivariate linear regression analyses, the dependent variables (nicotine and cotinine concentrations) were evaluated according to the sociodemographic data, participant characteristics, clinical and mental conditions, e-cigarette device features, and use patterns (Appendix A, respectively).

The nicotine concentration was significantly influenced exclusively by the duration since last e-cigarette consumption; participants who reported use within 30 min and within 1 h prior to sample collection had significantly higher nicotine mean concentrations compared to those who had used e-cigarettes more than 48 h prior (Appendix A), with no significant difference between the 30 min and 1 h groups (*p* = 0.274). Regarding the cotinine concentration, the variables that remained significant in the statistical model (*p* < 0.001) were last consumption, recharge/purchase frequency, and perception of addiction; participants who recharged/purchased their e-cigarettes daily, weekly, every two weeks, and monthly showed higher mean cotinine concentrations compared with those who did so occasionally, and participants who perceived that they experienced a mild, moderate, or severe addiction showed higher mean cotinine concentrations compared with those who reported experiencing no addiction (Appendix A), with no significant differences among addiction categories (*p* = 0.077).

The nicotine and cotinine concentrations in participants’ oral fluid were also evaluated according to participants’ knowledge of the nicotine content present in e-cigarettes, the forms of nicotine encountered, and their concentrations (Appendix A). For both nicotine and cotinine, significant differences were observed for knowledge of nicotine content and nicotine form.

The nicotine and cotinine concentrations for participants who reported being unaware of the presence of nicotine in e-cigarettes were similar to those who think e-cigarettes do not contain nicotine, both lower than those for participants completely aware of e-cigarettes’ nicotine content (Appendix A). A toxicological analysis showed that 55.2% of the samples from participants who thought e-cigarettes do not contain nicotine (N = 29) had detectable concentrations of the substance. Moreover, the nicotine and cotinine concentrations were higher for participants who reported using nicotine salt e-cigarettes compared with those reporting the use of nicotine free-base and nicotine-free e-cigarettes, as well as those who were unsure about the nicotine content. These lower-concentration groups exhibited similar mean concentrations (Appendix A).

Some participants exhibited very high nicotine levels (>1000 ng/mL) without corresponding increases in cotinine. Possible explanations include individual variability in nicotine metabolism, the shorter half-life of nicotine relative to cotinine, the saturation of metabolic pathways at high exposure levels, or occasional use characterized by intense consumption over a brief time span.

### 3.5. Participants’ Perception of Nicotine Addiction

Table 3 presents the sociodemographic data, participant characteristics, clinical and smoking history, e-cigarette use patterns, and nicotine and cotinine concentrations according to the participants’ perception of addiction.

Significant differences (*p* < 0.005) were observed for smoking history, consumption duration, recharge/purchase frequency, last consumption, and mean nicotine and cotinine concentrations. No statistical differences were observed with respect to gender, age, age range, body mass index, and presence of anxiety or depression.

In the group of participants who perceived no nicotine addiction, their mean nicotine and cotinine concentrations were lower. In the moderate and severe addiction groups, the proportion of former smokers, participants who last consumed e-cigarettes recently, and those who had been vaping for over 4 years were higher when compared with the proportion of participants who perceived that they did not have an addiction. In the severe addiction group, daily e-cigarette recharge/purchase frequency was higher when compared with the group who perceived that they did not have an addiction.

### 3.6. Subgroup Analysis of E-Cigarette Users with Nicotine Concentrations Above 400 ng/mL

Sociodemographic data, smoking history, e-cigarette use patterns, perception of addiction, and cotinine concentrations were evaluated in a subgroup of e-cigarette users with nicotine concentrations of 400 ng/mL or higher (Appendix A), which corresponds to nicotine levels typically found in individuals who smoke over 20 cigarettes per day.

Among the 376 participants with measurable nicotine concentrations, 49 exhibited levels exceeding 400 ng/mL, with significantly higher prevalence of devices based on nicotine salts, leading to significantly higher cotinine levels. Of these 49 participants, 15 showed nicotine levels above 1000 ng/mL; the highest level registered was 4557 ng/mL, and the mean level was 2400 ng/mL, which was six times the threshold of 400 ng/mL.

### 3.7. Participant Perception of Health and Social Impact, Addiction, Associated Risks, Attempts to Quit Vaping, Future Perspectives, and Government Regulations

Participants’ perception of impact on health status and social interactions, knowledge of risks and environmental consequences, perception of nicotine addiction, exposure to risk, attempts to quit vaping, future perspectives on e-cigarette use, and opinions on government regulations were evaluated according to smoking history (Appendix A).

Distinct distributions were observed for perception of impact on health status, knowledge of risks, perception of addiction, and opinion on government regulations with respect to smoking history. In the former smoker group, a higher proportion was observed for perceived health improvement. Conversely, in the never smoker group, the proportion of participants indicating no change in health impact was higher. However, almost 25% of the participants in both groups perceived that there was health deterioration.

More former smokers showed some knowledge of the risks involved in vaping and reported a moderate or severe addiction compared with never smokers. A higher proportion of former smokers (compared with never smokers) disagreed with current regulations and believed that the regulated commercialization of e-cigarettes should be authorized by the government. Never smokers did not perceive that they had nicotine addiction and indicated uncertainty about current regulations when compared with former smokers.

No significant differences were observed between groups for the remaining variables (knowledge of environmental consequences, exposure to risk, social impact, attempts to quit vaping, and future perspectives on e-cigarette use).

## 4. Discussion

Although e-cigarette use among Brazilian young adults increased following the COVID-19 pandemic, data from national surveys indicate that its prevalence fluctuated between 2019 and 2023, reaching 2.5% in 2020 and rising slightly to 2.6% in 2024. The highest prevalence was consistently observed in the 18–24-year-old age group.

The concern over the highest prevalence among young adults has prompted a renewed discussion on the topic by ANVISA’s Collegiate Board of Directors Resolution (RDC) No. 46/2009., which prohibited the commercialization of e-cigarettes in Brazil [1]. In 2024, RDC #855/2024 [11] was published, maintaining the prohibition of the manufacturing, importation, commercialization, distribution, storage, transportation, and advertising of all electronic smoking devices. Consequently, all forms of importation are banned, including for personal use and in travelers’ hand luggage. This decision was strongly supported by Brazilian medical societies, including the Brazilian Society of Cardiology, which issued a position statement advocating for maintenance of the ban [12]. One of the ten justifications in this statement was the comparison of prevalence rates between countries where e-cigarettes are permitted and where they are prohibited [13].

Our research provides a profile of e-cigarette users in the state of São Paulo, revealing that these individuals tend to be young adults with middle- to upper-class backgrounds, access to education, and high earning incomes. As such, they are less likely to rely on the Brazilian public healthcare system [14] for the treatment of nicotine dependence or related clinical complications. This suggests that e-cigarette use in this population imposes a lower direct burden on the public health system compared with that in other regions or socioeconomic groups with more limited access to private healthcare.

Additionally, approximately 61.8% of users were found to be either completely unaware of or have limited knowledge regarding the nicotine content of the products they used. Alarmingly, among participants who claimed to know that the product did not contain nicotine, 55% were found to have measurable concentrations of nicotine and cotinine (Appendix A). These findings highlight the critical role of product composition and user awareness in determining e-cigarette biomarker concentrations.

Some authors argue that the use of e-cigarettes by experienced users might, at most, be equivalent to the consumption of a pack of cigarettes [15]. However, our real-world study reveals alarming findings. The nicotine concentrations among e-cigarette users reached unprecedented levels compared with those of conventional cigarette smokers. Approximately 13% (N = 49) of users exhibited nicotine levels exceeding 400 ng/mL, with 15 individuals recording concentrations above 1000 ng/mL. In this subset of individuals with extreme nicotine exposure, the average nicotine concentration was six times higher than 400 ng/mL, a concentration typically observed in heavy conventional smokers [16]. Interestingly, in this subgroup, no linear relationship was observed with cotinine concentration, which was significantly higher than in participants with nicotine levels below 400 ng/mL, although not in the same proportion. Cotinine, the primary metabolite of nicotine, serves as a robust and reliable biomarker for nicotine exposure. Approximately 75–80% of nicotine is metabolized into cotinine [17], which has a longer half-life (16–20 h) and may partially explain the disproportionate nicotine/cotinine ratios observed, and its stability in biological fluids makes it an essential tool for evaluating nicotine consumption. Previous studies have demonstrated that the cotinine concentrations in e-cigarette users are often comparable with those in traditional cigarette smokers, indicating significant nicotine exposure from vaping [18,19,20,21]. However, none of these studies simultaneously evaluated the nicotine and cotinine levels in e-cigarette users.

We understand that this condition of extremely high nicotine levels, as well as cotinine levels, although not in the same proportion, may be attributed to the most used type of nicotine, nicotine salts. The dynamics of nicotine and cotinine in vapers using nicotine salt formulations have garnered significant scientific interest due to the distinctive pharmacokinetic properties of nicotine salts compared with those of traditional free-base nicotine. Nicotine salts, commonly found in pod-based e-cigarettes, particularly disposable devices, offer a smoother inhalation experience and enable the delivery of higher nicotine concentrations without the harsh throat irritation associated with the higher pH of nicotine free-base formulations [22]. This may also occur in users who have never smoked before or have only recently started using the product occasionally and, therefore, lack an adjusted capacity for nicotine metabolism.

These findings emphasize the risks associated with electronic nicotine delivery systems, particularly those using nicotine salts [23]. High nicotine concentrations not only increase the potential for addiction [24] but also raise serious concerns about toxicological impact at such elevated doses [19].

These levels surpass those typically reported in conventional tobacco users, reinforcing the urgent need for international attention to nicotine salts and their use in unregulated markets [4].

Moreover, studies have indicated that vapers often self-regulate nicotine intake, maintaining stable salivary cotinine levels even when reducing nicotine concentrations in e-liquids [25]. This behavior suggests that users adjust their vaping patterns to achieve consistent nicotine intake, highlighting the adaptability of consumption patterns. Variability in nicotine and cotinine levels can be attributed to factors such as e-liquid nicotine concentration, frequency of vaping, device type, and individual metabolic differences [26].

Notably, we observed no correlation between salivary nicotine concentrations and participants’ age or reported duration of e-cigarette use. This finding supports the idea that acute exposure is more strongly driven by current behaviors—such as device type, nicotine concentration, and time since last use—rather than long-term patterns. The pharmacokinetic properties of nicotine salts may further contribute to this dynamic, particularly among newer or infrequent users.

In adolescents, the cotinine concentrations among e-cigarette users have been reported to exceed those observed in conventional cigarette smokers, raising serious concerns about nicotine dependence and its associated health risks [27]. These risks include cognitive impairment, increased susceptibility to nicotine addiction, and potential impacts on brain development and pregnancy outcomes. Additionally, acute cardiovascular effects, such as elevated blood pressure and increased heart rate, have also been reported in young users.

In summary, vapers using nicotine salt formulations are exposed to high levels of nicotine, which is reflected in their nicotine and cotinine levels. The ability to self-titrate nicotine intake, combined with the pharmacokinetic advantages of nicotine salts, results in significant nicotine exposure that parallels or exceeds that of traditional cigarette smoking. This underscores the need for continued research into the health implications of vaping, particularly among vulnerable populations such as adolescents.

Although Brazil maintains a national ban on electronic smoking devices (RDC 46/2009 and RDC 855/2024), enforcement remains limited due to resource constraints and the focus of penalties on sellers rather than users. However, the prohibition itself facilitates identification and control efforts, as banned products are more easily recognized and classified as illicit. Despite this, most participants reported purchasing devices in tobacco shops or online, indicating a persistent market fueled by weak oversight. These findings highlight the need to strengthen inspections, surveillance, and interagency coordination to effectively limit illegal access and distribution.

Nicotine and cotinine concentrations are dependent on (a) device type—advanced devices (such as mods or pods) allow for greater control over temperature and power, leading to increased nicotine absorption (Appendix A); (b) e-liquid nicotine levels—liquids with high nicotine concentrations (including nicotine salts) are associated with higher salivary nicotine levels; (c) duration since last consumption—nicotine levels drop quickly within hours (half-life of 2 h), whereas cotinine, with a longer half-life (16–20 h), remains detectable for a longer period; (d) recharge/purchase frequency for disposable devices—frequent users tend to have higher levels of nicotine and cotinine in their saliva, as well as higher degrees of nicotine dependence; and (e) perception of nicotine addiction—dependence was significantly related to nicotine and cotinine concentrations (Table 3).

### Study Limitations

Our study could be affected by selection bias due to its use of convenience sampling. The participants were limited to adults aged ≥18 years, which excluded a critical population of e-cigarette users, particularly adolescents in middle and high schools. The inclusion criteria included exclusive e-cigarette users who were former smokers and had quit smoking at least 1 month prior to this study; smoking cessation was confirmed through their carbon monoxide concentration in exhaled air, where those with 4 ppm or higher were excluded. Passive smoking exposure was not evaluated. Marijuana consumption via electronic smoking devices was excluded based solely on self-reports, whereas smoked marijuana was captured through measurements of carbon monoxide concentration in exhaled air. Finally, 41 samples were lost due to insufficient oral fluid volume, and retesting was not possible. In addition, the questionnaire used was not formally validated psychometrically, which may affect the reliability and reproducibility of some subjective measures. Finally, our recruitment strategy may have introduced additional selection bias. Because the participants were recruited in public places such as nightlife venues, the sample may overrepresent younger, more socially active individuals with higher socioeconomic statuses. However, the use of random health inspection schedules helped to mitigate bias by ensuring a broad geographic and social representation.

## 5. Conclusions

Most e-cigarette users in the study sample recruited from the state of São Paulo, Brazil, belong to the middle or upper classes, are predominantly White, and have access to education and high earning incomes.

Salivary nicotine and cotinine levels have proven to be robust and non-invasive biomarkers for assessing smoking status and nicotine dependence. These biomarkers are critical in both clinical and epidemiological research, providing reliable measures for monitoring nicotine exposure and its associated health impacts.

Disposable devices containing nicotine salts were the most commonly used products among the participants. A strong correlation was observed between perceived level of nicotine dependence and salivary concentrations of nicotine and cotinine. Higher perceived dependence was associated with elevated biomarker levels, as well as higher rates of moderate to severe addiction, particularly among individuals with longer durations of use.

High salivary concentrations of nicotine appeared to be independent of duration of e-cigarette use, smoking history, and age. This highlights the significant risks of nicotine intoxication, even in younger individuals or first-time users.

These findings underscore the urgent need for public health initiatives to address the accessibility of these devices and to raise awareness about the rapid onset of nicotine dependence and its potential links to mental health disorders. These results offer strong empirical support for ANVISA’s continued prohibition of e-cigarettes and highlight the need for enhanced enforcement and public awareness campaigns to curb widespread illicit access.

## Figures and Tables

**Figure 1 ijerph-22-00960-f001:**
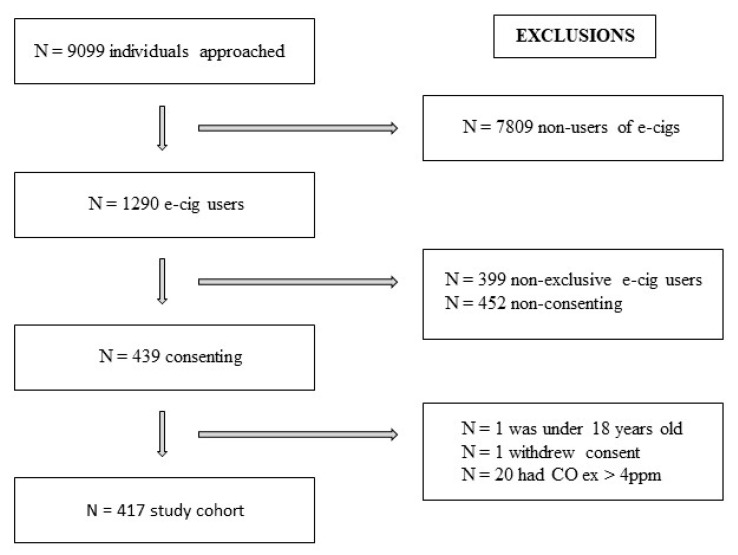
Flowchart depicting participant selection and the corresponding exclusion criteria applied at each stage of data collection.

**Figure 2 ijerph-22-00960-f002:**
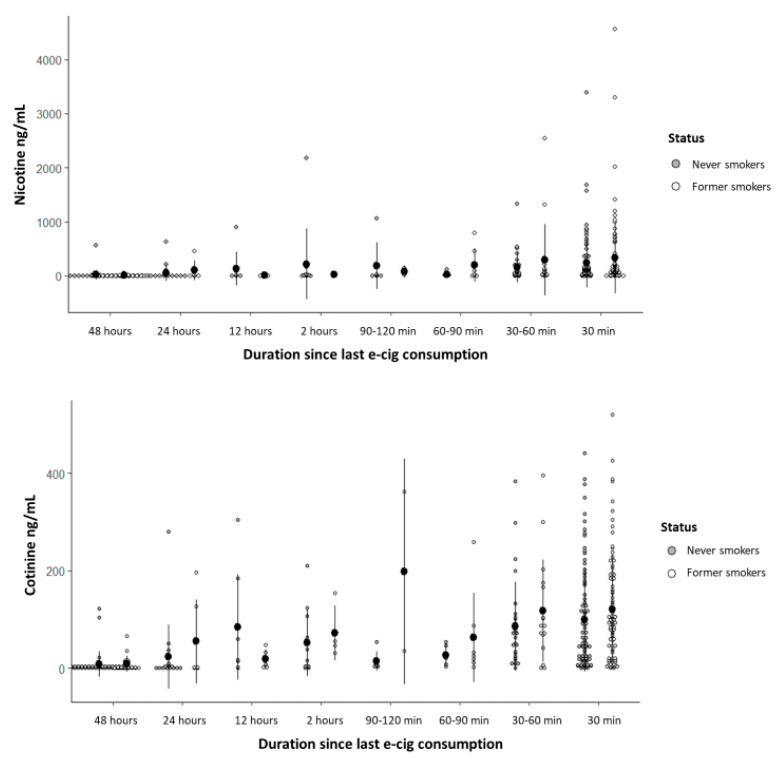
Dot plot for nicotine and cotinine concentrations with respect to duration since last e-cigarette consumption in former and never smokers. Data are presented as mean ± standard deviation.

**Figure 3 ijerph-22-00960-f003:**
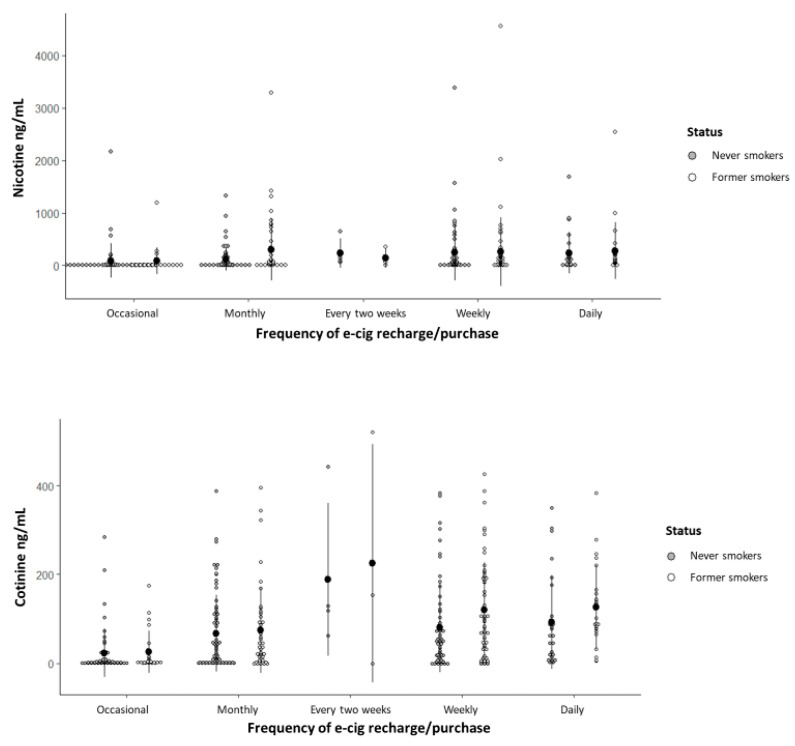
Dot plot for nicotine and cotinine concentrations with respect to e-cigarette recharge/purchase frequency in former and never smokers. Data are presented as mean ± standard deviation.

**Table 1 ijerph-22-00960-t001:** Participant sociodemographic profiles, physical characteristics, and health conditions according to smoking history.

	Overall	Smoking History		*p* *
		Never Smokers	Former Smokers	
Gender, n (%)	N = 403	N = 230	N = 173	0.021 ^a^
Male	211 (52.4)	109 (47.4)	102 (59.0)	
Female	192 (47.6)	121 (52.6)	71 (41.0)	
Age, years	N = 403	N = 230	N = 173	<0.001 ^b^
Mean ± SD	28.0 ± 9.4	25.5 ± 7.8	31.3 ± 10.3	
Age range, n (%)	N = 403	N = 230	N = 173	<0.001 ^a^
≤25 years	207 (51.4)	147 (63.9)	60 (34.7)	
26–35 years	111 (27.5)	54 (23.5)	57 (32.9)	
36–45 years	63 (15.6)	22 (9.6)	41 (23.7)	
≥46 years	22 (5.5)	7 (3.0)	15 (8.7)	
Race, n (%)	N = 399	N = 228	N = 171	0.094 ^c^
White	273 (68.4)	157 (68.9)	116 (67.8)	
Black	30 (7.5)	11 (4.8)	19 (11.1)	
Asian	7 (1.8)	5 (2.2)	2 (1.2)	
Multiracial	89 (22.3)	55 (24.1)	34 (19.9)	
Years spent in education, n (%)	N = 399	N = 229	N = 170	0.180 ^a^
≤9 years	14 (3.5)	6 (2.6)	8 (4.7)	
10–15 years	169 (42.4)	105 (45.9)	64 (37.6)	
≥16 years	215 (53.9)	117 (51.1)	98 (57.6)	
Monthly earning income, n (%)	N = 352	N = 202	N = 150	0.125 ^a^
≤BRL 1000.00	3 (0.9)	2 (1.0)	1 (0.7)	
BRL 1001.00–2500.00	54 (15.3)	36 (17.8)	18 (12.0)	
BRL 2501.00–5000.00	98 (27.8)	62 (30.7)	36 (24.0)	
BRL 5001.00–10,000.00	89 (25.3)	50 (24.8)	39 (26.0)	
≥BRL 10,000.00	108 (30.7)	52 (25.7)	56 (37.3)	
Mean body mass index (BMI), kg/m^2^	21.81 ± 4.58	20.8 ± 4.21	22.15 ± 4.9	
Presence of clinical conditions, n (%)	N = 400	N = 228	N = 172	0.895 ^c^
Yes	71 (17.8)	42 (18.4)	29 (16.9)	
Allergy	17 (23.9)	11 (26.2)	6 (20.7)	0.593 ^a^
Asthma	16 (22.5)	6 (14.3)	10 (34.5)	0.045 ^a^
Hypertension	12 (16.9)	5 (11.9)	7 (24.1)	0.209 ^c^
Diabetes mellitus	5 (7.0)	2 (4.8)	3 (10.3)	0.393 ^c^
Other	26 (36.6)	18 (42.9)	8 (27.6)	0.189 ^a^
Presence of mental disorders, n (%)	N = 398	N = 227	N = 171	0.536 ^c^
Yes	125 (31.4)	67 (29.5)	58 (33.9)	
Anxiety	94 (75.2)	48 (71.6)	46 (79.3)	0.322 ^a^
Depression	39 (31.2)	16 (23.9)	23 (39.7)	0.058 ^a^
Bipolar disorder	8 (6.4)	5 (7.5)	3 (5.2)	0.724 ^a^
Personality disorder	4 (3.2)	3 (4.5)	1 (1.7)	0.623 ^c^
Schizophrenia	1 (0.8)	1 (1.5)	0 (0.0)	1.000 ^c^
Other	6 (4.8)	5 (7.5)	1 (1.7)	0.215 ^c^

N: total number of participants considered in the analysis; n: number of participants within a subgroup; SD: standard deviation; IQR: interquartile range. BRL: Brazilian reais, the official currency of Brazil. * *p*-values indicate the statistical significance of the results of ^a^ chi-square test, ^b^ Mann–Whitney test, and ^c^ Fisher’s exact test.

**Table 2 ijerph-22-00960-t002:** Information on e-cigarette consumption patterns, product characteristics, purchases and expenses, and nicotine and cotinine concentrations in oral fluid, according to smoking history.

	Overall	Smoking History		*p* *
		Never Smokers	Former Smokers	
Introduction to e-cigarettes, n (%)	N = 399	N = 227	N = 172	
By friends and/or family	264 (66.2)	156 (68.7)	108 (62.8)	0.215 ^a^
Self-interest	74 (18.5)	32 (14.1)	42 (24.4)	0.009 ^a^
Through advertising	65 (16.3)	40 (17.6)	25 (14.5)	0.408 ^a^
Other	14 (3.5)	10 (4.4)	4 (2.3)	0.264 ^a^
Reasons for use, n (%)	N = 402	N = 230	N = 172	
Out of curiosity	163 (40.5)	125 (54.3)	38 (22.1)	<0.001 ^a^
Influence of friends and/or family	116 (28.9)	86 (37.4)	30 (17.4)	<0.001 ^a^
Tobacco smoking cessation	71 (17.7)	0 (0)	71 (40.1)	<0.001 ^a^
Alternative to traditional cigarettes	59 (14.7)	16 (7.0)	43 (25.0)	<0.001 ^a^
Advertising	16 (4.0)	8 (3.5)	8 (4.7)	0.552 ^a^
Other	65 (16.2)	38 (16.5)	27 (15.7)	0.824 ^a^
Product type, n (%)	N = 401	N = 229	N = 172	
Disposable	297 (74.1)	175 (76.4)	122 (70.9)	0.215 ^a^
Rechargeable	100 (24.9)	55 (24.0)	45 (26.2)	0.623 ^a^
E-liquid vaporization	42 (10.5)	21 (9.2)	21 (12.2)	0.325 ^a^
Heated tobacco	3 (0.7)	1 (0.4)	2 (1.2)	0.579 ^b^
Consumption duration, n (%)	N = 396	N = 226	N = 170	0.013 ^a^
≤1 year	154 (38.8)	102 (44.7)	52 (30.8)	
2–3 years	172 (43.3)	92 (40.4)	80 (47.3)	
≥4 years	71 (17.9)	34 (14.9)	37 (21.9)	
Last consumption, n (%)	N = 397	N = 228	N = 169	0.116 ^b^
30 min ago	197 (49.6)	99 (43.4)	98 (58.0)	
31–60 min ago	51 (12.8)	33 (14.5)	18 (10.7)	
61–90 min ago	16 (4.0)	7 (3.1)	9 (5.3)	
90–120 min ago	9 (2.3)	6 (2.6)	3 (1.8)	
>2 h ago	16 (4.0)	12 (5.3)	4 (2.4)	
~12 h ago	16 (4.0)	10 (4.4)	6 (3.6)	
~24 h ago	25 (6.3)	18 (7.9)	7 (4.1)	
~48 h ago	64 (16.1)	40 (17.5)	24 (14.2)	
Last week	2 (0.5)	2 (0.9)	0 (0.0)	
Last month	1 (0.3)	1 (0.4)	0 (0.0)	
Place of purchase, n (%)	N = 398	N = 227	N = 171	
Internet	102 (25.6)	54 (23.8)	48 (28.1)	0.333 ^a^
Tobacco shop	278 (69.8)	161 (70.9)	117 (68.4)	0.590 ^a^
Abroad	9 (2.3)	2 (0.9)	7 (4.1)	0.042 ^b^
Gift from friends and/or family	36 (9.0)	22 (9.7)	14 (8.2)	0.604 ^a^
Recharge/purchase frequency, n (%)	N = 396	N = 226	N = 170	0.029 ^a^
Daily	54 (13.6)	29 (12.8)	25 (14.7)	
Weekly	128 (32.3)	60 (26.5)	68 (40.0)	
Every two weeks	9 (2.3)	5 (2.2)	4 (2.4)	
Monthly	127 (32.1)	79 (35.0)	48 (28.2)	
Occasional	78 (19.7)	53 (23.5)	25 (14.7)	
Monthly expenses, n (%)	N = 395	N = 225	N = 170	<0.001 ^b^
≤BRL 50.00	57 (14.4)	36 (16.0)	21 (12.4)	
BRL 51.00–100.00	109 (27.6)	77 (34.2)	32 (18.8)	
BRL 101.00–300.00	162 (41.0)	88 (39.1)	74 (43.5)	
BRL 301.00–500.00	47 (11.9)	14 (6.2)	33 (19.4)	
BRL 501.00–1000.00	16 (4.1)	8 (3.6)	8 (4.7)	
≥BRL 1000.00	2 (0.5)	0 (0.0)	2 (1.2)	
None (provided by friends and/or family)	2 (0.5)	2 (0.9)	0 (0.0)	
Knowledge of nicotine content, n (%)	N = 400	N = 228	N = 172	0.091 ^a^
No knowledge	215 (53.8)	130 (57.0)	85 (49.4)	
Aware of nicotine content	153 (38.3)	77 (33.8)	76 (44.2)	
Unaware of nicotine content	32 (8.0)	21 (9.2)	11 (6.4)	
Nicotine form, n (%)	N = 400	N = 228	N = 172	
Nicotine salt	110 (27.5)	58 (25.4)	52 (30.2)	0.288 ^a^
Nicotine free-base products	50 (12.5)	22 (9.6)	28 (16.3)	0.047 ^a^
No knowledge	217 (54.3)	132 (57.9)	85 (49.4)	0.092 ^a^
No nicotine	29 (7.3)	20 (8.8)	9 (5.2)	0.177 ^a^
Nicotine salt concentration, n (%)	N = 110	N = 58	N = 52	0.929 ^b^
3 mg/mL	1 (0.9)	1 (1.7)	0 (0.0)	
20 mg/mL	18 (16.4)	10 (17.2)	8 (15.4)	
35 mg/mL	24 (21.8)	11 (19.0)	13 (25.0)	
50 mg/mL	58 (52.7)	31 (53.4)	27 (51.9)	
Unknown	9 (8.2)	5 (8.6)	4 (7.7)	
Nicotine free-base concentration, n (%)	N = 50	N = 22	N = 28	0.193 ^b^
3 mg/mL	30 (60.0)	15 (68.2)	15 (53.6)	
6 mg/mL	16 (32.0)	5 (22.7)	11 (39.3)	
12 mg/mL	1 (2.0)	1 (4.5)	0 (0.0)	
18 mg/mL	0 (0.0)	0 (0.0)	0 (0.0)	
>20 mg/mL	1 (2.0)	1 (4.5)	0 (0.0)	
Unknown	2 (4.0)	0 (0.0)	2 (7.1)	
Nicotine, ng/mL	N = 362	N = 208	N = 154	0.023 ^c^
Mean ± SD	194.90 ± 466.49	158.99 ± 373.86	243.39 ± 565.88	
Median (IQR)	38.50 (6.00–148.50)	25.00 (5.00–137.75)	62.00 (9.75–178.50)	
Cotinine, ng/mL	N = 362	N = 208	N = 154	0.001 ^c^
Mean ± SD	76.90 ± 98.70	64.52 ± 91.85	93.61 ± 105.27	
Median (IQR)	36.00 (3.00–111.00)	20.00 (2.25–91.00)	67.00 (8.00–140.25)	

N: total number of participants considered in the analysis; n: number of participants within a subgroup; SD: standard deviation; IQR: interquartile range. BRL: Brazilian reais, the official currency of Brazil. * *p*-values indicate the statistical significance according to a ^a^ chi-square test, ^b^ Fisher’s exact test, and ^c^ Mann–Whitney test. Knowledge of nicotine content: “No knowledge” refers to users who did not know the nicotine concentration. “Unaware” refers to those who believed that the product contained no nicotine. “Aware” indicates users who knew the nicotine concentration of the product.

**Table 3 ijerph-22-00960-t003:** Sociodemographic data, participant characteristics, clinical and smoking history, e-cigarette use patterns, and nicotine and cotinine concentrations according to the participants’ perception of addiction.

	Participants’ Perception of Nicotine Addiction					*p* *
	None	Mild	Moderate	Severe	Unsure	
Gender, n (%)	N = 154	N = 88	N = 94	N = 70	N = 9	0.883 ^a^
Male	83 (53.9)	45 (51.1)	47 (50.0)	36 (51.4)	6 (66.7)	
Female	71 (46.1)	43 (48.9)	47 (50.0)	34 (48.6)	3 (33.3)	
Age, years	N = 154	N = 88	N = 94	N = 70	N = 9	0.458 ^b^
Mean ± SD	27.01 ± 9.48	28.16 ± 8.75	28.15 ± 10.06	28.49 ± 9.03	29.44 ± 9.89	
Median (IQR)	24.00 (20.00–30.25)	26.00 (21.00–34.75)	24.00 (21.00–32.25)	26.00 (21.00–34.25)	25.00 (21.50–40.00)	
Age range, n (%)	N = 154	N = 88	N = 94	N = 70	N = 9	0.806 ^c^
≤25 years	87 (56.5)	41 (46.6)	52 (55.3)	34 (48.6)	5 (55.6)	
26–35 years	39 (25.3)	27 (30.7)	21 (22.3)	21 (30.0)	1 (11.1)	
36–45 years	22 (14.3)	16 (18.2)	14 (14.9)	11 (15.7)	2 (22.2)	
≥46 years	6 (3.9)	4 (4.5)	7 (7.4)	4 (5.7)	1 (11.1)	
Body mass index, kg/m^2^	N = 154	N = 87	N = 94	N = 69	N = 9	0.207 ^c^
Mean ± SD	21.78 ± 4.41	21.16 ± 4.09	22.13 ± 5.29	21.89 ± 4.60	24.36 ± 3.60	
Median (IQR)	20.96 (18.56–25.03)	20.59 (18.13–23.88)	21.16 (18.99–24.39)	21.25 (18.37–24.92)	24.17 (21.42–27.79)	
Smoking history, n (%)	N = 150	N = 86	N = 91	N = 66	N = 8	<0.001 ^a^
Never smokers	107 (71.3)	50 (58.1)	37 (40.7)	30 (45.5)	4 (50.0)	
Former smokers	43 (28.7)	36 (41.9)	54 (59.3)	36 (54.5)	4 (50.0)	
Consumption duration, n (%)	N = 151	N = 87	N = 93	N = 70	N = 8	<0.001 ^a^
≤1 year	82 (54.3)	42 (48.3)	22 (23.7)	10 (14.3)	2 (25.0)	
2–3 years	58 (38.4)	36 (41.4)	45 (48.4)	34 (48.6)	4 (50.0)	
≥4 years	11 (7.3)	9 (10.3)	26 (28.0)	26 (37.1)	2 (25.0)	
Recharge/purchase frequency, n (%)	N = 152	N = 87	N = 92	N = 69	N = 8	<0.001 ^a^
Daily	11 (7.2)	13 (14.9)	17 (18.5)	16 (23.2)	1 (12.5)	
Frequent **	86 (56.6)	59 (67.8)	64 (69.6)	49 (71.0)	3 (37.5)	
Occasional	55 (36.2)	15 (17.2)	11 (12.0)	4 (5.8)	4 (50.0)	
Last consumption, n (%)	N = 152	N = 88	N = 92	N = 70	N = 7	<0.001 ^a^
Recent (<24 h ago)	105 (69.1)	76 (86.4)	85 (92.4)	70 (100.0)	5 (71.4)	
Not recent (>24 h ago)	47 (30.9)	12 (13.6)	7 (7.6)	0 (0.0)	2 (28.6)	
Presence of anxiety or depression, n (%)	N = 146	N = 88	N = 94	N = 68	N = 9	0.252 ^a^
No	111 (76.0)	66 (75.0)	64 (68.1)	44 (64.7)	8 (88.9)	
Yes	35 (24.0)	22 (25.0)	30 (31.9)	24 (35.3)	1 (11.1)	
Nicotine, ng/mL	N = 139	N = 79	N = 82	N = 65	N = 9	<0.001 ^b^
Mean ± SD	127.68 ± 310.46	219.89 ± 493.33	201.55 ± 532.08	324.48 ± 589.12	26.67 ± 41.29	
Median (IQR)	10.00 (0.00–80.00)	44.00 (9.00–142.00)	69.50 (14.75–190.25)	99.00 (38.00–362.00)	0.00 (0.00–62.50)	
Cotinine, ng/mL	N = 139	N = 79	N = 82	N = 65	N = 9	<0.001 ^b^
Mean ± SD	37.72 ± 74.43	81.28 ± 92.61	103.17 ± 102.23	127.15 ± 112.81	45.44 ± 96.75	
Median (IQR)	4.00 (0.00–35.00)	46.00 (15.00–107.00)	72.50 (19.75–155.75)	96.00 (41.00–194.00)	6.00 (2.00–45.00)	

N: total number of participants considered in the analysis; n: number of participants within a subgroup; SD: standard deviation; IQR: interquartile range. * *p*-values indicate the statistical significance of the results of ^a^ chi-square test, ^b^ Mann–Whitney test, and ^c^ Fisher’s exact test. ** Frequent means weekly, every two weeks, or monthly.

## Data Availability

The original contributions presented in this study are included in this article/its Appendix A. Further inquiries can be directed to the corresponding author(s).

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
