# Peer review of "Nicotine Dependence in a Banned Market: Biomarker Evidence from E-Cigarette Users in São Paulo, Brazil"

_ijerph, 2025, doi:10.3390/ijerph22060960_

Round 1

Reviewer 1 Report

Comments and Suggestions for Authors

This study investigated characteristics of exclusive e-cigarette users among adults in Sao Paulo, Brazil, where e-cigarettes are banned. The article is well organized and easy to read. The research questions were well-defined and relevant to the tobacco-control field. The methods were appropriate and valid to address the research questions. Broadening the sample to include all e-cigarette users would have improved our understanding of the extent of e-cigarette use, but that would have made it more difficult to isolate the biological effects of e-cigarette use (i.e., nicotine levels). The statistical methods were appropriate, and the authors provided a clear rationale for their statistical choices. The data presentation was clear and sufficiently detailed. The discussion was balanced and hewed close to the data. A significant strength of the study is its inclusion of biomarkers of nicotine and cotinine levels, which provided insights into the elevated addiction and nicotine exposure risks among users of devices with nicotine salt delivery systems. Discussing the current enforcement strategies and limitations would be helpful, as it appears that products are mostly purchased in tobacco shops. In addition, it would be useful to know if smokers in Brazil can access other types of nicotine replacement products to control their cigarette smoking addiction (e.g., nicotine lozenges, gum, etc.) and reduce the harms associated with smoking combustible cigarettes. In summary, this is an excellent contribution to the literature that advances our understanding of the use and potential harms of e-cigarettes despite the government prohibition of use.

Author Response

Comment 1: Discussing the current enforcement strategies and limitations would be helpful, as it appears that products are mostly purchased in tobacco shops.

Response 1: Thank you for this suggestion. We have added a brief discussion on current enforcement strategies and challenges in Brazil. Specifically, we mentioned that despite the national ban, enforcement is inconsistent, with limited inspection coverage and low penalties, which enables the continued presence of e-cigarette products in retail settings, particularly in tobacco shops. This addition was included in the penultimate paragraph of the Discussion section.

Comment 2: It would be useful to know if smokers in Brazil can access other types of nicotine replacement products to control their cigarette smoking addiction (e.g., nicotine lozenges, gum, etc.) and reduce the harms associated with smoking combustible cigarettes.

Response 2: We appreciate this important point. In Brazil, nicotine replacement therapies (NRTs) such as patches and gum are legally available in pharmacies, with or without a prescription, and are also offered free of charge through the smoking cessation program of the Unified Health System (SUS). However, the present study did not assess the use of NRTs among e-cigarette users. It is important to note that individuals who were using any form of nicotine replacement therapy were excluded from the study, as this was an exclusion criterion.

Comment 3: This is an excellent contribution to the literature that advances our understanding of the use and potential harms of e-cigarettes despite the government prohibition of use.

Response 3: We sincerely thank the reviewer for this generous and encouraging feedback.

Reviewer 2 Report

Comments and Suggestions for Authors

what was the goal of comparing never/naive smokers to former smokers here?  Perhaps a short explanation in your methods would be helpful. Was not clear on how the comparison was helpful to your main points. 

Comments on the Quality of English Language

If The term “naive smoker” means they have not ever smoked a cigarette, then the term should be “never smoker” in English.

Author Response

Comment 1: What was the goal of comparing never/naive smokers to former smokers here? Perhaps a short explanation in your methods would be helpful.

Response 1: Thank you for your question. We clarified in the Methods section (2.1. Study design and procedures) that the comparison between never smokers and former smokers was included to explore whether prior tobacco smoking history influenced nicotine exposure, product use patterns, or perceived addiction in exclusive e-cigarette users. This stratification also allowed us to evaluate differences in behavioral and biomarker outcomes related to previous nicotine exposure.

Comment 2: If the term “naive smoker” means they have not ever smoked a cigarette, then the term should be “never smoker” in English.

Response 2: Thank you for pointing this out. We replaced all occurrences of “naïve smoker” with “never smoker” to align with the correct terminology in English-language publications.

Reviewer 3 Report

Comments and Suggestions for Authors

The aim of this study was “to characterize exclusive e-cigarette users in terms of user sociodemographic data, e-cigarette consumption patterns, perception of addiction and health risks, device accessibility, health and environmental impact, government regulations, attempts to quit vaping, future perspectives and social implications of e-cigarette use, and to correlate these variables with nicotine and cotinine concentrations in oral fluid samples”.

The results were relevant; however, I had some observations.

  1. Lines 125-126 Please correct Mass Body Index (MBI); the correct thing is Body Mass Index (BMI).
  2. Line 126 Describe in detail the criteria for “former smokers, naïve smokers”
  3. Describe in detail the classification used for "participants' perception of addiction". What criteria did each of the subgroups presented in Table 3 have to meet?
  4. Was the questionnaire previously validated? If so, we would appreciate mentioning its psychometric properties (validity, reliability).
  5. In addition to exhaled CO, what other criteria were used to exclude dual users?
  6. What were the limit of detection (LOD) and limit of quantification (LOQ) for nicotine and cotinine in the LC-MS/MS technique used?
  7. In some cases with extremely high levels of nicotine (>1000 ng/mL), no proportional increase in cotinine was observed. How do you explain this pharmacokinetic dissociation?
  8. Were the models adjusted for the presence of mental disorders, considering their possible influence on the perception of addiction?
  9. How do you interpret the fact that salivary nicotine concentrations do not correlate with duration of use or age?
  10. Add the meaning of BRL, it is presented as an abbreviation in Table 1.
  11. The dot plots in Figure 2 are not clear due to the color coding used. I suggest adding a bullet indicating which corresponds to Naive and Former Smokers, respectively.
  12. Do you have estimates of the potential impact of selection bias due to the recruitment environment?
  13. According to other articles in the group, the correct name is “Surveillance System for Risk and Protective Factors for Chronic Diseases by Telephone Survey (Vigitel)”; please check.

Author Response

Comment 1: Lines 125–126: Please correct "Mass Body Index (MBI)"; the correct term is Body Mass Index (BMI).

Response 1: Thank you for identifying this error. We corrected the terminology to "Body Mass Index (BMI)" throughout the manuscript.

Comment 2: Line 126: Describe in detail the criteria for “former smokers” and “naïve smokers”.

Response 2: We revised the Methods section to include detailed definitions:

  • “Never smokers” were defined as individuals who reported never having smoked tobacco products in their lifetime.
  • “Former smokers” were defined as individuals who reported having quit smoking tobacco products at least one month prior to the study, with exhaled carbon monoxide levels ≤3 ppm to confirm abstinence.

Comment 3: Describe in detail the classification used for "participants' perception of addiction".

Response 3: We updated the Methods section to explain that participants were asked to self-rate their perception of nicotine addiction as: none, mild, moderate, severe, unsure.

This was a subjective self-assessment with no pre-defined criteria, as is standard in perception-based survey instruments.

Comment 4: Was the questionnaire previously validated? If so, please mention its psychometric properties.

Response 4: We clarified in the Methods that the questionnaire was developed by the research team and based on prior national surveys and peer-reviewed literature. It has not been formally validated psychometrically, and this was acknowledged as a limitation in the revised Discussion.

Comment 5: Besides exhaled CO, what other criteria were used to exclude dual users?

Response 5: We specified in the Methods that exclusion of dual users relied on self-report of tobacco and marijuana use and was complemented by exhaled CO measurement to identify possible cigarette or combusted marijuana use. Participants reporting dual use or with CO ≥4 ppm were excluded.

Comment 6: What were the LOD and LOQ for nicotine and cotinine in the LC-MS/MS technique used?

Response 6: We added in the Methods the following information:

  • Limit of Detection (LOD): 1 ng/mL for both nicotine and cotinine
  • Limit of Quantification (LOQ): 5 ng/mL for nicotine and 1 ng/mL for cotinine

Comment 7: In some cases with extremely high nicotine levels (>1000 ng/mL), no proportional increase in cotinine was observed. How do you explain this pharmacokinetic dissociation?

Response 7: We expanded the Discussion to explore this phenomenon. We suggest that the disparity may be related to individual variability in nicotine metabolism, recentness of consumption (nicotine has a shorter half-life than cotinine), and saturation of metabolic pathways in cases of extremely high exposure.

Comment 8: Were the models adjusted for mental disorders, considering their possible influence on addiction perception?

Response 8: We thank the reviewer for this observation. We conducted a cross-tabulation analysis between self-perceived nicotine dependence (none, mild, moderate, severe) and the presence or absence of mental disorders. The results showed no statistically significant differences between the groups. Therefore, mental disorders were not included as an adjustment variable in the final models.

Comment 9: How do you interpret the fact that salivary nicotine concentrations do not correlate with duration of use or age?

Response 9: Thank you for this important observation. We have now clarified this interpretation in the Discussion section. The lack of correlation between salivary nicotine concentrations and either age or duration of use suggests that current use patterns (e.g., device type, nicotine formulation, frequency of use) have a greater influence on acute exposure than long-term history. This is particularly relevant in the context of nicotine salts, which enable rapid absorption and high bioavailability. These findings highlight that factors such as the time of last use, frequency of vaping, and device characteristics better explain biomarker levels than demographic or historical variables.

Comment 10: Add the meaning of BRL, it is presented as an abbreviation in Table 1.

Response 10: We added a footnote to Table 1 indicating that BRL = Brazilian Reais, the official currency of Brazil.

Comment 11: The dot plots in Figure 2 are not clear due to the color coding used.

Response 11: We respectfully acknowledge the reviewer’s comment. However, we have opted to maintain the current color coding, as it ensures consistency with the visual identity of the other figures in the manuscript and remains distinguishable when printed in grayscale

Comment 12: Do you have estimates of potential selection bias due to recruitment environment?

Response 12: We appreciate the reviewer’s important question. Although we did not calculate formal estimates of selection bias, we acknowledge that our sampling and recruitment in public venues may have favored the inclusion of more socially active or health-conscious individuals, particularly those from urban and university-affiliated settings. This may have led to overrepresentation of younger participants with higher education and income levels. We have added a sentence to the Discussion section to reflect this limitation and its potential impact on generalizability.

Comment 13: Correct the name to “Surveillance System for Risk and Protective Factors for Chronic Diseases by Telephone Survey (Vigitel)”

Response 13: Thank you. We corrected the name of the Vigitel system throughout the manuscript to its full and proper English title.

Reviewer 4 Report

Comments and Suggestions for Authors

The authors conducted a population and (nicotine consumption) biomarker-based cross-sectional study of exclusive e-cigarette users (people who do not also smoke nicotine) in six municipalities in São Paulo, Brazil in 2024. The goal was to gather data that can be used to develop/improve public health policies and intervention strategies to decrease nicotine vaping.

Their aim was “to characterize exclusive e-cigarette users in terms of user sociodemographic data, e-cigarette consumption patterns, perception of addiction and health risks, device accessibility, health and environmental impact, government regulations, attempts to quit vaping, future perspectives and social implications of e-cigarette use, and to correlate these variables with nicotine and cotinine concentrations in oral fluid samples.” A wealth of information was obtained.

The survey addressed an important and timely subject, especially as more is being learned about the adverse health effects of vaping and use by those 18-24 remains concerning. A limitation, and area for additional study, is that only people 18 years of age and older were included. Prevalence rates are much lower than in countries where vaping is legal, offsetting potential negative effects of unregulated products. The extraordinarily high nicotine levels in some users highlight the risks of lack of regulation, and the need for public education. Given the mounting data on the high risks of dual use, it is extremely worrisome that almost half (399) the e-cig users sampled were non-exclusive e-cig users.

As concluded by the researchers: “These findings underscore the urgent need for public health initiatives to address the accessibility of these devices and to raise awareness about the rapid onset of nicotine dependence and its potential links to mental health disorders. These results offer strong empirical support for ANVISA’s continued prohibition of e-cigarettes, and highlight the need for enhanced enforcement and public awareness campaigns to curb widespread illicit access.”

How well is the ban on e-cigarettes enforced? What are the penalties? Are users at risk of punishment?

Starting line 63 “the use of these products continues to rise, especially among youth and young adults… perceived increase in consumption was observed following COVID-19 pandemics, particularly among young adults.” These statements are not supported by Vigitel and Covitel data between 2019 and April 2023. Line 72: “prevalence of e-cigarette use increased from 2.3% in 2019 to 2.5% in 2020”; then they went to 2.2%. This has been partially addressed in the Discussion. Additionally, those numbers are for all those 18 years or older. Recommend including prevalence rates for those 18-24 (6.4% and 7.0% for 2019 and 2020 per Vigitel), and the most recent prevalence data, differentiated by age. Recommend including prevalence data for youth 13-17 years. Consider including comparable prevalence rates from other countries.

Line 88 – “…high nicotine concentrations may indicate hazardous usage patterns or improper exposure”. What is meant by “improper exposure”?

Line 140 “participants were questioned if they put themselves at risk in order to acquire e-cigarettes”; recommend briefly giving examples of the type of risk.

Methods: Nicotine replacement therapy, and oral forms of nicotine (tobacco, pouches, etc.) can add to cotinine levels; do you know if any were being used?

Figure 1. What does to mean that 9099 people were “approached”? Did surveyors walk up to them and ask if they were willing to be included? That 14% (1290) were e-cig users suggests that the effort to sample populations at risk worked.

Table 1: There is no gender information for 14 of 417 in the study cohort. Were other options given? No category includes more than 403. Did participants decline to answer?

Table 2.

Introduction to e-cigs, and reasons for use: What does “self-interest” mean? What is the difference between “tobacco smoking cessation” and “alternative to traditional cigarettes”?

Knowledge of nicotine: Does “no knowledge” mean that they did not know there was nicotine in the e-cigs, or that they had no idea how much nicotine? How does that differ from “unaware of nicotine content”?

Line 216. “Based on smoking history, distinct distributions (p < 0.05) were observed for the following parameters...” Recommend briefly describing those distributions.

Line 402. “In adolescents, cotinine concentrations among e-cigarette users have been reported to exceed those observed in conventional cigarette smokers, raising serious concerns about nicotine dependence and associated health risks.” Consider giving examples of the health risks (effects on developing brains, risks related to pregnancy, etc.).

Author Response

Comment 1: How well is the ban on e-cigarettes enforced? What are the penalties? Are users at risk of punishment?

Response 1: We thank the reviewer for this important comment. The issues raised regarding enforcement, penalties, and user risk were incorporated into the revised Discussion section, following a similar suggestion from another reviewer. We clarified that enforcement of the national ban is limited due to resource constraints, that legal penalties target primarily manufacturers and sellers, and that individual users are not criminalized. These points were included in the broader discussion of regulatory challenges and the persistence of the illicit market.

Comment 2: Lines 63–72: Statements about increased use post-COVID are not supported by Vigitel/Covitel data. Include age-specific prevalence rates and youth data.

Response 2: We thank the reviewer for this observation. In the Introduction, we mentioned an increase in e-cigarette use in Brazil following the COVID-19 pandemic and cited VIGITEL data showing a rise in prevalence from 2019 to 2020. In the Discussion, we acknowledged a subsequent fluctuation, with prevalence decreasing to 2.1% in 2023. However, more recent data from VIGITEL 2024—reported but not yet published as of May 29, 2025—indicate a prevalence of 2.6% among individuals aged 18 and older. We focused our analysis on this age group, as it matches the population included in our study.

Comment 3: Line 88: What is meant by “improper exposure”?

Response 3: We replaced “improper exposure” with “unintentional high-dose exposure, including among users unaware of nicotine concentration in their devices.”

Comment 4: Line 140: Provide examples of risks users reported to acquire e-cigarettes.

Response 4: This refers to whether the participant felt at risk of experiencing violence or danger in order to obtain e-cigarettes, given their illegal status. The concept of “risk to acquire e-cigarettes” includes situations such as traveling to unsafe neighborhoods, purchasing from unlicensed vendors, or using false identification.

Comment 5: Could other forms of nicotine use have contributed to cotinine levels (e.g., pouches, NRT)?

Response 5: We added to the exclusion criteria section that, although participants were screened for tobacco and marijuana smoking, the use of other nicotine products—such as oral nicotine or NRTs—was not systematically assessed and may have influenced biomarker levels in a small number of cases. Exclusive e-cigarette use was an inclusion criterion, and the use of other nicotine products, including conventional, NRTs, hand-rolled, or roll-your-own cigarettes, was part of the exclusion criteria.

Comment 6: Clarify what is meant by "approached" in Figure 1.

Response 6: We thank the reviewer for the observation. In Figure 1, “approached” refers to the 9,099 individuals who were approached by field teams and invited to participate during public health inspections.

Comment 7: Table 1: Why is there missing gender data? Were other gender options offered?

Response 7: We clarified in the Methods that gender was self-reported. Participants could choose “male,” “female,” or “prefer not to say.”

Comment 8: Clarify what “self-interest” means in Table 2.

Response 8: “self-interest” refers to individuals who sought out e-cigarettes independently, without influence from others or advertising.

Comment 9: Distinguish between “tobacco smoking cessation” and “alternative to traditional cigarettes.”

Response 9: “Tobacco smoking cessation” is using e-cigarettes as a strategy to quit smoking. “Alternative to traditional cigarettes” is continued use of nicotine through vaping rather than conventional smoking.

Comment 10: Clarify the categories for “Knowledge of nicotine content.”

Response 10: We have clarified the categories in the footnotes of Table 2. “No knowledge” refers to users who do not know the nicotine concentration. “Unaware of nicotine content” refers to those who believe the product contains no nicotine. “Aware” indicates users who know the nicotine concentration of the product.

Comment 11: “Based on smoking history, distinct distributions (p < 0.05) were observed for the following parameters...” Recommend briefly describing those distributions.

Response 11:  As we can see in Table 2, the reasons for using e-cigarettes differ between those who have never smoked and former smokers

Never smoker

Ex-smoker

Reasons for use, n (%)

N=402

N=230

N=172

Out of curiosity

163 (40.5)

125 (54.3)

38 (22.1)

<0.001a

Influence of friends and/or family

116 (28.9)

86 (37.4)

30 (17.4)

<0.001a

Tobacco smoking cessation

71 (17.7)

0 (0)

71 (40.1)

<0.001a

Alternative to traditional cigarettes

59 (14.7)

16 (7.0)

43 (25.0)

<0.001a

Comment 12: Line 402: When citing health risks to adolescents, include examples.

Response 12: We have expanded the text to include specific examples of health risks associated with e-cigarette use among adolescents, such as cognitive impairment, increased susceptibility to addiction, and potential impacts on brain development and pregnancy outcomes. We also added that acute cardiovascular effects—such as elevated blood pressure and increased heart rate—have been observed in young users.

Reviewer 5 Report

Comments and Suggestions for Authors

Dear editor and authors,

I have read the manuscript "Nicotine Dependence in a Banned Market: Biomarker Evidence from E-Cigarette Users in São Paulo, Brazil".

The combination of salivary biomarkers and questionnaires is interesting. The aim of the project seems to be finding correlations. However, how this aim supports reaching a specific goal in public health is not clearly described in the abstract or introduction. I would advise to more clearly describe this aim. A big question that I have is: what is the additional value of the biomarkers? Would future studies only need saliva biomarkers and no more questionnaires or would the authors advise a combination of both? I am missing clear recommendations on this matter. Also, from the results I see that most comparisons are done between naïve and former smokers. Why these two groups are compared is not clear for me from the introduction/aim.

One of the initial questions I had when reading this manuscript was the last time use of an e-cigarette before taking the saliva sample. Indeed, the authors described the nicotine concentration strongly depends on last time of e-cigarette use (L252). What is not fully clear to me is how this affects the search for other correlations? Was/can a correction (be) performed for this?

The chemical analytical method was validated according to the authors, but all information on the validation parameters (LOD, LOQ etc.) is missing. Please provide this information in manuscript or supplement.

Some interesting findings are the correlations of cotinine with recharge/purchase frequency and perception of addiction. Also the finding that quite some users thought not having nicotine in their e-cigarettes, which did contain nicotine I find a relevant finding. I would be nice to know what brand varieties were used by these users and investigate nicotine contents on labels and/or by chemical analysis of the e-liquid. This in order to make sure that the detected nicotine was derived from this product and also to get more information on whether labels are being read by users or for instance that products are mislabeled. The detection of high nicotine concentrations (~20 cig/day) is also an important finding, have more researches observed this in exclusive e-cigarette users?

The conclusions section summarizes some results and then there are some recommendations (e.g. accessibility of the devices). How these recommendations are linked to the results is not clearly described.

Although multiple aspects of this research might be relevant, I am missing a clear aim and a clear connection between the researchers findings and their recommendations. Based on my findings I would advise to revise the manuscript (major revisions).

Other comments:

  • P1: L38: “Population-based, …” sentence is missing a verb.
  • P2: L 60-61: “The characterization … potential health impacts.” Sentence doesn’t read well. Perhaps rephrasing the last part. Health impacts are not a ‘characteristic’ of an e-cigarette user, but a consequence.
  • P2: L62-L63: “In Brazil.. Nevertheless …” my suggestion is to replace these sentences with L66-L68 “Although … adults”. Since this seems a repetition of the same thing. And if another way of solving this issue is chosen, please put the reference on increased e-cigarette use when first mentioned.
  • P2: L73: 2.3% to 2.5% I don’t think this is a significant increase. Further down in the discussion this is also better described as ‘stabilized’, since in 2023 prevalence is at 2.1% (L336). These small fluctuations (2.3% - 2.5% - 2.1%) seem a pretty stable trend to me.
  • P2: L80: I would argue that biomarkers are essential to answer the question of nicotine dependence in e-cigarette users not seeking smoking cessation treatment. Aren’t questionnaires a good tool (even as stand-alone) to answer this question?
  • P3 L116-118: please specify at what point the oral fluid sample was taken. Before/after questionnaire? In P3 L130 the question is described on when they last used it, but I would like to see a clearer sentence on last use before sampling the saliva.
  • P3: L148: the reference (10) does not refer to the LC-MS/MS method, but a protocol for validation.
  • P3 L154: what fragments/ions were used as qualifier/quantifier? What was the used LC method (isocratic/gradient solvents etc.)?
  • P3 L158: how was the method validated? A reference (10) for a validation protocol is mentioned at the beginning of this section, but have all these parameters been determined? And what were the results in matter of limits of detection/quantification, reproducibility, recovery etc. Please specify (also possible to put in Supplement).
  • P6 L212: Table 2. Nicotine and cotinine concentrations in e-liquid or saliva? Please specify in table.
  • P10 L 256: please specify which groups (the 30min and 1h groups?).
  • P13 L372-P14 L374: I would advise to already mention the half-life of cotinine here (now only mentioned in L416). Might this partially explain the different ratios between users with more and less than 400 ng/mL nicotine?
  • P14 L380-383: please add reference that supports the different pharmacokinetics of nicotine as free-base or salt.
  • P15: L442: “A strong correlation was observed between perceived level of nicotine dependence and salivary concentrations of nicotine and cotinine.” I thought this was only true for cotinine, based on the results (P10 L257-258), but perhaps it is not clear to me what is meant, please clarify.
  • P15: L451-453 “These results…” this sentence is a repetition of the preceding sentence (L449-451)

Author Response

Comment 1: A big question that I have is: what is the additional value of the biomarkers?

Response 1: Thank you for raising this important question. In the context of studying electronic cigarette use, we believe that incorporating biomarkers alongside questionnaires is essential. The use of two biomarkers—nicotine (to assess acute exposure) and cotinine (to evaluate regular consumption)—provided valuable insights into participants’ usage patterns and helped validate self-reported data. This dual approach enhanced the accuracy and depth of our assessment of e-cigarette consumption behavior.

Comment 2: One of the initial questions I had when reading this manuscript was the last time use of an e-cigarette before taking the saliva sample. Indeed, the authors described the nicotine concentration strongly depends on last time of e-cigarette use (L252). What is not fully clear to me is how this affects the search for other correlations? Was/can a correction (be) performed for this?

Response 2: Thank you for this thoughtful observation. Indeed, the timing of last e-cigarette use is crucial when interpreting nicotine concentrations, given its short half-life—approximately 50% is metabolized within 1–2 hours. To address this, we collected self-reported data on the time since last use and used cotinine, a more stable metabolite with a longer half-life, as our primary biomarker to assess regular use patterns. While we acknowledge that variability in the timing of nicotine intake may affect acute-level correlations, the use of cotinine helped mitigate this issue in most analyses.

Comment 3: The chemical analytical method was validated according to the authors, but all information on the validation parameters (LOD, LOQ etc.) is missing. Please provide this information in manuscript or supplement.

Response 3: Thank you for this observation. The LOD and LOQ values have already been included in the manuscript in response to another reviewer. The method was fully developed at the Laboratory of Toxicology of the University of São Paulo Medical School, and although it has not yet been published, its validation followed standard practices for analytical method validation in forensic toxicology. A reference to the international guidelines used has also been added to the manuscript.

Comment 4: Some interesting findings are the correlations of cotinine with recharge/purchase frequency and perception of addiction. Also the finding that quite some users thought not having nicotine in their e-cigarettes, which did contain nicotine I find a relevant finding. I would be nice to know what brand varieties were used by these users and investigate nicotine contents on labels and/or by chemical analysis of the e-liquid. This in order to make sure that the detected nicotine was derived from this product and also to get more information on whether labels are being read by users or for instance that products are mislabeled.

Response 4: Thank you for highlighting this important point. Unfortunately, most users in our study did not report paying close attention to product labels. Many of them reported using e-cigarettes shared with friends or family members, which may contribute to a sense of safety and reduced concern about the product's content. Participants who demonstrated greater awareness were primarily former smokers who had transitioned to e-cigarettes; this subgroup tended to be more informed about the nicotine content and brand of the products they used.

Comment 5: The conclusions section summarizes some results and then there are some recommendations (e.g. accessibility of the devices). How these recommendations are linked to the results is not clearly described.

Response 5: Thank you for your comment. In Brazil, the sale of e-cigarettes is currently prohibited by federal legislation. Our findings demonstrate high levels of nicotine exposure and a strong potential for dependence among users, including those who believed their devices did not contain nicotine. These results reinforce concerns about the addictive nature of e-cigarettes and provide evidence to support existing regulatory restrictions, particularly the importance of not increasing accessibility through legalization.

Comment 6: Although multiple aspects of this research might be relevant, I am missing a clear aim and a clear connection between the researchers findings and their recommendations. Based on my findings I would advise to revise the manuscript (major revisions).

Response 6: Thank you very much for your thoughtful feedback. We appreciate your observation regarding the need for a clearer statement of the study’s aim and a stronger connection between our findings and recommendations. Based on your comments and those of the other reviewers, we have carefully revised the manuscript to enhance clarity and explicitly articulate the study objectives. Additionally, we have strengthened the discussion to better link our key findings with the policy and public health recommendations. We believe these changes improve the overall coherence and impact of the manuscript and hope they address your concerns. 

Comment 7: P1: L38: “Population-based, …” sentence is missing a verb.

Response 7: Thank you for pointing that out. The sentence has been revised to include the missing verb and now reads more clearly in the manuscript.

Comment 8: P2: L62-L63: “In Brazil… Nevertheless …” my suggestion is to replace these sentences with L66-L68 “Although… adults”. Since this seems a repetition of the same thing. And if another way of solving this issue is chosen, please put the reference on increased e-cigarette use when first mentioned. P2: L73: 2.3% to 2.5% I don’t think this is a significant increase. Further down in the discussion this is also better described as ‘stabilized’, since in 2023 prevalence is at 2.1% (L336). These small fluctuations (2.3% - 2.5% - 2.1%) seem a pretty stable trend to me.

Response 8: Thank you for your suggestion. The paragraph has been revised to eliminate redundancy and improve clarity. We chose to retain part of the original content but restructured the paragraph, updated it with the most recent 2024 data, and repositioned the reference on increased e-cigarette use to its first mention, as recommended.

Comment 9: P2: L80: I would argue that biomarkers are essential to answer the question of nicotine dependence in e-cigarette users not seeking smoking cessation treatment. Aren’t questionnaires a good tool (even as stand-alone) to answer this question? 

Response 9: Thank you for raising this important point. While standardized questionnaires are indeed valuable tools for assessing nicotine dependence, we believe that the inclusion of biomarkers such as nicotine and cotinine levels provides an objective measure of recent exposure. This biochemical data complements self-reported information and allows for a more comprehensive understanding of both the extent of exposure and physiological dependence, especially in populations not actively seeking cessation. Therefore, combining both approaches enhances the accuracy and depth of the assessment.

Comment 10: P3 L116-118: please specify at what point the oral fluid sample was taken. Before/after questionnaire? In P3 L130 the question is described on when they last used it, but I would like to see a clearer sentence on last use before sampling the saliva.

Response 10: Thank you for your observation. Briefly, participants first signed the informed consent form, completed the questionnaires, and then underwent exhaled carbon monoxide (CO) measurement. If the CO level was below 4 ppm, the oral fluid sample was collected immediately afterward. Additionally, the questionnaire included a specific item on the timing of last nicotine or e-cigarette use prior to the saliva sampling, allowing us to contextualize the biomarker results.

Comment 11: P3: L148: the reference (10) does not refer to the LC-MS/MS method, but a protocol for validation. P3 L154: what fragments/ions were used as qualifier/quantifier? What was the used LC method (isocratic/gradient solvents etc.)? P3 L158: how was the method validated? A reference (10) for a validation protocol is mentioned at the beginning of this section, but have all these parameters been determined? And what were the results in matter of limits of detection/quantification, reproducibility, recovery etc. Please specify (also possible to put in Supplement).

Response 11: Thank you for your detailed and constructive comment. We agree that this section required further clarification. In response, we have uploaded a Supplementary Material to provide more specific information about the LC-MS/MS method, including the chromatographic conditions (mobile phases, gradient elution), the MRM transitions used for nicotine and cotinine (qualifier and quantifier ions), and the internal standards applied. We also clarified that the method was validated in-house following the ASB Standard Practices for Method Validation in Forensic Toxicology (reference 10), and we now provide detailed validation parameters—such as limits of detection (LOD), limits of quantification (LOQ), precision, accuracy, recovery, and matrix effects. These results confirm that the method is suitable for detecting and quantifying nicotine and cotinine in oral fluid.

Comment 12: P6 L212: Table 2. Nicotine and cotinine concentrations in e-liquid or saliva? Please specify in table.

Response 12: Thank you for pointing this out. We have updated the legend of Table 2 to clearly state that the nicotine and cotinine concentrations refer to oral fluid samples.

Comment 13: P10 L 256: please specify which groups (the 30min and 1h groups?).

Response 13: Thank you for the observation. We have revised the sentence to explicitly indicate that the comparison refers to the groups who reported e-cigarette use within 30 minutes and within 1 hour prior to sample collection.

Comment 14: P13 L372-P14 L374: I would advise to already mention the half-life of cotinine here (now only mentioned in L416). Might this partially explain the different ratios between users with more and less than 400 ng/mL nicotine?

Response 14: Thank you for this valuable suggestion. We have incorporated information on the half-life of cotinine earlier in the Discussion section, as recommended. We also acknowledged that differences in the nicotine/cotinine ratio may be partially explained by the longer half-life of cotinine, which tends to accumulate in the body even after nicotine levels decline.

Comment 15: P14 L380-383: please add reference that supports the different pharmacokinetics of nicotine as free-base or salt.

Response 15: Thank you for the comment. The information regarding the different pharmacokinetics of nicotine as free-base or salt is supported by references 22 and 24, which are already cited in the relevant paragraphs.

Comment 16: P15: L442: “A strong correlation was observed between perceived level of nicotine dependence and salivary concentrations of nicotine and cotinine.” I thought this was only true for cotinine, based on the results (P10 L257-258), but perhaps it is not clear to me what is meant, please clarify.

Response 16: Thank you for your question. Indeed, individuals with high perceived nicotine dependence are typically regular nicotine consumers, so elevated levels of both cotinine and nicotine are expected. However, individuals who do not report dependence may still use nicotine abusively but sporadically and in large quantities. This pattern can result in high nicotine levels without correspondingly elevated cotinine levels, since cotinine accumulates with regular use. We have clarified this point in the manuscript to avoid any confusion.

Comment 17: P15: L451-453 “These results…” this sentence is a repetition of the preceding sentence (L449-451)

Response 17: Thank you for your suggestion. We chose to reiterate this point because we believe it is important to emphasize the need for public policies that restrict access to the product and for educational campaigns. It is worth highlighting that ANVISA’s recent public policy has been recognized by the WHO as an effective measure. We have maintained this phrasing to reinforce these critical aspects.

Round 2

Reviewer 3 Report

Comments and Suggestions for Authors

The authors addressed most of the comments. Only the image editing remained.